# Effect of Fluid Flow on the Corrosion Performance of as-Cast and Heat-Treated Nickel Aluminum Bronze Alloy (UNS C95800) in Saline Solution

**Hamid Reza Jafari** [1,2] , **Ali Davoodi** [1,*] and **Saman Hosseinpour** [1,3,*]

1   Materials and Metallurgical Engineering Department, Ferdowsi University of Mashhad, Mashhad 9177948974, Iran; hamidrezajafari2010@yahoo.com
2   New Materials Technology and Processing Research Center, Department of Metallurgical Engineering, Neyshabur Branch, Islamic Azad University, Neyshabur 9319975853, Iran
3   Institute of Particle Technology (LFG), Friedrich-Alexander-Universität-Erlangen-Nürnberg (FAU), Cauerstraße 4, 91058 Erlangen, Germany
*   Correspondence: a.davoodi@um.ac.ir (A.D.); saman.hosseinpour@fau.de (S.H.)

**Abstract:** In this work, the corrosion behavior and surface reactivity of as-cast and heat-treated nickel aluminum bronze casting alloy (UNS C95800) in 3.5 wt% NaCl solution is investigated under stagnant and flow conditions. Increasing flow rate conditions are simulated using a rotating disk electrode from 0 to 9000 revolutions per minute (rpm). Optical micrographs confirm the decrease in the phase fraction of corrosion-sensitive β phase in the microstructure of C95800 after annealing, which, in turn, enhances the corrosion resistance of the alloy. Electrochemical studies including open circuit potentiometry, potentiodynamic polarization, and electrochemical impedance spectroscopy are performed to assess the effect of flow rate and heat treatment on the corrosion of samples at 25 and 40 °C in 3.5 wt% NaCl solution. For both as-cast and heat-treated samples, increasing the flow rate (i.e., electrode rotating rate) linearly reduces the corrosion resistance, indicating that the metal dissolution rate is significantly affected by hydrodynamic flow. Increasing the solution temperature negatively impacts the corrosion behavior of the as-cast and heat-treated samples at all flow conditions.

**Keywords:** nickel aluminum bronze; annealing; flow dynamics; rotating disc electrode; corrosion; electrochemistry

## 1. Introduction

Copper alloys containing nickel and aluminum are referred to as nickel-aluminum bronze (NAB) and often contain 5–11 wt% aluminum [1]. Due to their mechanical and corrosion resistance properties, NABs have found many industrial applications, especially in the marine environment [2,3]. For instance, a large fraction of giant valves and blades in marine applications are composed of casted NABs, which meet the toughness and electrochemical resistance requirements in the corrosive saline environment. The mechanical properties of castings and wrought NAB alloys increase with their aluminum content [4]. The mechanical and corrosion behaviors of NABs are largely dependent on its chemical composition as well as the post-casting heat treatment of these alloys; many studies have been performed to assess the mechanical strength [5,6], wear resistance [7], fatigue [8], and corrosion behavior [9] of NABs as a function of their chemical composition and microstructure. UNS C95800 alloy with 8.5–9.5 wt% Al and 3.5–4.5 wt% Ni is one of the frequently used NABs for the fabrication of large casting parts in contact with seawater, such as propellers. The microstructure of the as-cast C95800 alloy is comprised of alpha (α) and beta (β) phases as well as a small fraction of intermetallic kappa (κ) phases, the fraction and distribution of which are highly dependent on the cooling rate after casting and the subsequent heat treatment. The phase distribution and microstructure of C95800 affect the

corrosion performance of the alloy during its exposure to certain corrosive environments. Degradation of NAB casting alloys for naval applications is associated with the preferential corrosion of the $\kappa_{III}$ lamellar intermetallic phase, which, in turn, reduces the mechanical strength of the alloy [10]. As such, many studies have focused on optimizing the heat treatment of the alloy to improve its corrosion resistance [1,5]. For instance, Yang et al. showed that the addition of Ni to NAB alloys changes the content of phases in favor of $\kappa$ phase formation, which increases both corrosion resistance and yield strengths [11]. In the marine environment where the water salinity is a determining factor in electrochemical interactions and the corrosion of metallic parts, the growth and stability of the corrosion products or protective layer are the key factors determining the performance and service life of the metals and alloys. Therefore, the corrosion resistance of NABs is correlated with the nature, formation, and properties of the protective interfacial layer, which contains mixed oxides and hydroxides of Cu, Al, and Ni [12]. Schüssler and Exner estimated the thickness of this oxide layer as being in the order of ca. 1000 nm [12]. According to Song et al., this protective layer is composed of inner CuxO and AlxO and outer copper oxide and hydroxide [13]. In stagnant conditions, the rate of the anodic and cathodic reactions consequently reduces the corrosion rate of the samples [3,5,12]. Nevertheless, in most cases, these surfaces in marine conditions are subjected to turbulent flow, which can alter the chemical and electrochemical equilibria at the metal/protective layer/corrosive environment interfaces or can mechanically affect the stability of these layers. Therefore, apart from considering the chemical composition and microstructure of the substrate, for the proper assessment of the corrosion protection that is offered by such interfacial layers, the experimental conditions should be chosen to closely represent the actual exposure conditions, and the impact of the fluid flow cannot be neglected.

The rotating disc electrode (RDE) method can be used to evaluate of the effect of fluid flow on the electrochemical interactions under benign and turbulent flows [14], during which the electrochemical responses of the substrates are considered a measure of their corrosion performance. RDE has been successfully used to assess the corrosion resistance of metals and alloys under well-defined hydrodynamic conditions [15–17], which better enables the determination of the effects of flowing parameters on the corrosion mechanisms compared to the conventional methods of solution agitation. For instance, Marco and Van der Biest observed that the formation rate of the layer composed of corrosion products on the surface of pure magnesium during polarization is reduced by the use of RDE [18]. High speed RDE was also used to study the simultaneous erosion and corrosion of copper-based alloys in sea water [19]. Ge et al. used a graphite RDE to conduct electrochemical measurements under a well-defined flow condition and verified the coexistence of $Eu^{3+}$ and $Eu^{2+}$ in molten LiCl-KCl salt, which has important implications for the development of molten salt reactors [20]. The aim of this study was thus to systematically assess the effect of flow rate on the corrosion resistance of the as-cast and annealed C95800 alloy in 3.5 wt% NaCl solution at two different solution temperatures (25 and 40 °C).

## 2. Materials and Methods

### 2.1. Samples

Rods of nickel aluminum bronze alloy with a diameter of 5 mm and length of 15 mm were cut from the as-cast samples. The chemical composition of the as-cast samples was determined using optical emission spectrometry and the results are provided in Table 1. The obtained chemical composition was consistent with the UNS C95800. Annealing heat treatment was performed according to ASTM B148 standard [21], in which the sample was heat-treated at 675 °C for 6 h followed by cooling down in air to room temperature.

**Table 1.** Chemical composition of the as-cast C95800 sample in wt%.

| Cu | Al | Cr | Si | Ni | Fe | Mn | P | Sn | Pb | Zn | S | Co |
|---|---|---|---|---|---|---|---|---|---|---|---|---|
| Base | 8.79 | 0.007 | 0.08 | 4.50 | 3.70 | 1.00 | 0.02 | 0.02 | <0.01 | 0.05 | 0.01 | <0.01 |

## 2.2. Microstructure Studies

To visualize the microstructure of the as-cast and heat-treated alloys, samples were wet-ground using successive grades of SiC paper up to 2000 grit. The samples were degreased using ethanol and then rinsed with distilled water and dried in air. Etchant solution containing 25 mL HCl, 8 g $FeCl_3$, and 100 mL distilled water was used to selectively etch the phase boundaries [22]. An optical microscope (model GX51, Olympus, Tokyo, Japan) was used to capture the optical micrographs. Phase fraction quantification was performed using image analysis (μgrain software V3.00, Mark-Henning, Germany), in accordance with ASTM E562 [23] and ASTM E1242 [24] standards.

## 2.3. Electrochemical Measurements

For electrochemical studies, the sample surface was polished with SiC paper up to 2000 grit, rinsed with a copious amount of ethanol, and dried in air. A potentiostat (model E10800 Compactstat, IVIUM, Eindhoven, The Netherlands) and RDE device (model AFMSRX, Pine Research Instrumentation, Durham, NC, USA) were coupled for electrochemical measurements in stagnant and hydrodynamic flow conditions.

The conventional three-electrode cell was used with as-cast or heat-treated C95800 as the working electrode, and saturated Ag/AgCl and platinum as the reference and counter electrodes, respectively. We used 250 mL of 3.5 wt% NaCl solution as the electrolyte for electrochemical studies. The distance between electrodes was controlled to minimize the IR drop and errors in the electrochemical measurements.

Electrochemical measurements included, in this order, the measurement of the open circuit potential (OCP) over time, electrochemical impedance spectroscopy (EIS), and potentiodynamic polarization (PDP). To evaluate the effect of fluid flow on the corrosion resistance of the samples, the RDE device was set to rotation speeds between 0 and 9000 rpm. Electrochemical measurements were performed on as-cast and heat-treated samples at two different solution temperatures (25 and 40 °C).

## 3. Results

### 3.1. Microscopic Investigations

Figure 1 depicts the microstructure of as-cast and heat-treated C95800 alloy at two different magnifications (200× and 1500×). The Widmanstätten morphology, which can be observed in Figure 1a,b, formed due to the slow cooling rate after casting of the C95800 alloy [25], which resulted in the formation of a considerable amount of metastable β phase in the microstructure of the alloy. The chemical composition of the β phase is close to $Cu_3Al$ [26] and acts as the anode in comparison with the more noble Cu-rich α phase in the electrochemical reactions. The substantial amount of defects in the β phase also negatively impacts its corrosion resistance [26]. After annealing heat treatment, as shown in Figure 1c,d, the β phase dissolved and the fraction ratio of the α phase further increased. As discussed later, this reduction in the fraction of the more sensitive β phase in the microstructure of the heat-treated C95800 resulted in the overall improvement in the corrosion resistance of the alloy.

As shown in Figure 2, the quantitative comparison of the phase fractions in the as-cast and heat-treated samples using image analysis indicated an almost 13% decrease in the β phase fraction (from >20% to ~7%) after the annealing heat treatment. The fraction of kappa phase in relation to the α and β phases may have been overestimated by the software due to the fine structure of the kappa phase. Nevertheless, the relative increase in the fraction of the kappa phase after heat treatment is clearly visible in Figure 1. As mentioned earlier, during the annealing heat treatment, the β phase dissolved while the fraction of κ phase almost doubled (from >33% to >66%). The κ phase mainly consists of $\kappa_I$, $\kappa_{II}$, $\kappa_{III}$, and $\kappa_{IV}$ small precipitates, which contain larger fractions of Ni and Al compared to the β phase [1,6]. The high content of Al in κ phase assists in the formation of the corrosion resistance film on the surface, whereas Ni, in general, increases the corrosion resistance of the alloy [6]. The more uniform distribution of Ni, Al, and Fe within the interfacial layer

and formation of corrosion resistance complexes such as NiAl after heat treatment results in the formation of a more uniform and protective layer with improved ionic and electrical resistance. Therefore, it can be expected that these changes in the microstructure of the C95800 after heat treatment would increase its corrosion resistance, which is consistent with our electrochemical observations (vide infra).

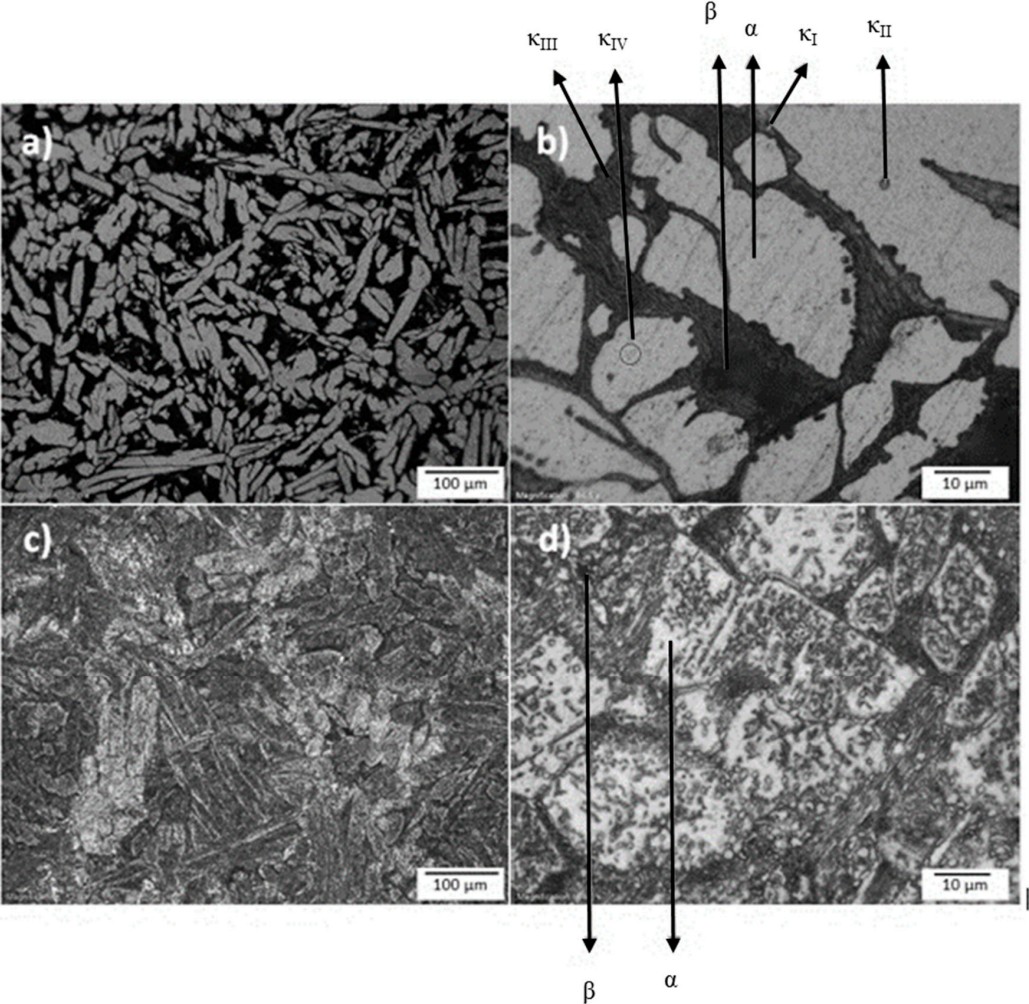

**Figure 1.** Microstructure of as-cast C95800 at (**a**) 200× and (**b**) 1500× and heat-treated C95800 sample at (**c**) 200× and (**d**) 1500×.

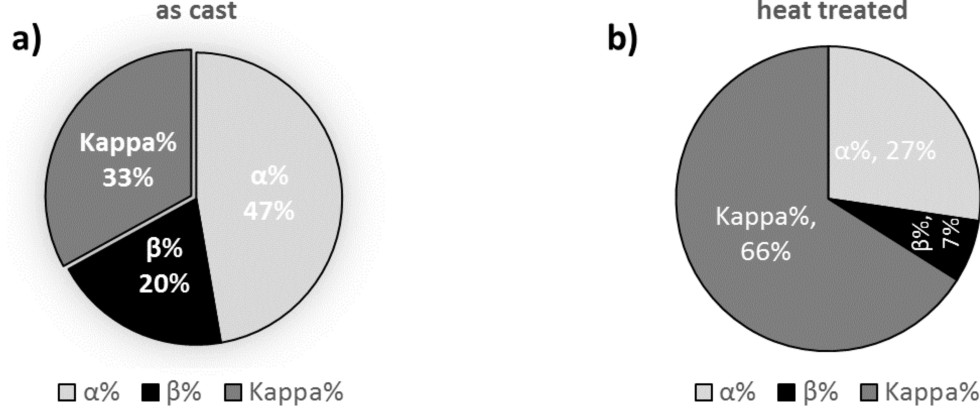

**Figure 2.** Quantitative phase fraction (%) in the (**a**) as-cast and (**b**) heat-treated C95800 alloy. The phase fractions were estimated using image analysis software.

### 3.2. Electrochemical Evaluations

3.2.1. OCP Measurements

Figure 3a,d depicts the changes in the OCP of the as-cast and heat-treated C95800 alloy with exposure time in 3.5 wt% NaCl at two different solution temperatures (25 and 40 °C) under the stagnant condition as well as RDE rotation speeds of 1000–9000 rpm. As as soon as the samples were brought into contact with the corrosive solution, the potentials changed toward more negative values. This shift in potential indicated the impact of flow on electrochemical reactions close to the sample surface, especially on accelerating the anodic reactions. Corrosion is accompanied by the diffusion of metal ions from the corroding surface toward the bulk solution and the diffusion of oxygen toward the surface, both of which are enhanced by the flow of solution. In stagnant conditions (denoted as static), the reduction in the OCP gradually continues until the end of the measurement (i.e., 1000 s). In contrast, when the samples were exposed to a fluid flow (e.g., using the RDE), these changes in the OCP were relatively quick initially and then slowed down to the minimum of ~1 mV/300 s, which is considered as equilibrium state at the interface between the alloy and solution. The time to reach the equilibrium state in all samples reduced as the rotation speed increased.

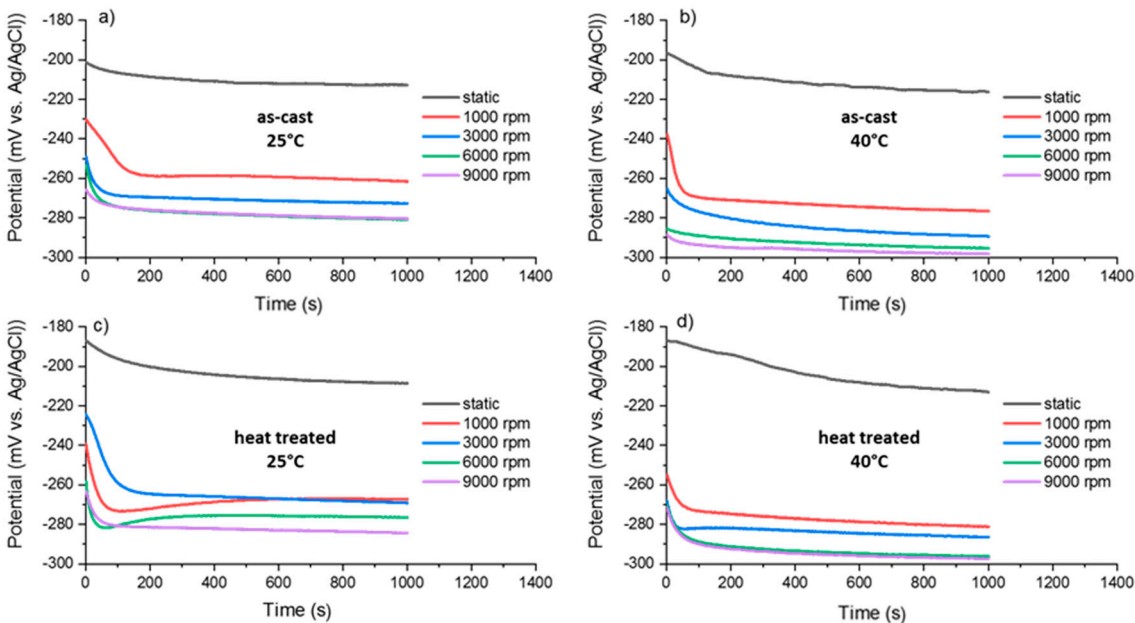

**Figure 3.** Open circuit potential variations for C95800 alloy as a function of immersion time in 3.5 wt% NaCl solution at different rotating disc electrode's rotation speeds for (**a**) as-cast sample at 25 °C, (**b**) as-cast sample at 40°C, (**c**) heat-treated sample at 25 °C, and (**d**) heat-treated sample at 40 °C.

The observed changes in the OCP indicated that the electrochemical reactions on the sample surfaces are mass-transfer-controlled and the rate of these reactions can be increased by increasing the rotation speed (or fluid flow rate). In other words, as the rotation speed increases, the transport of metal ions and corrosion products and access of the corrosive ions to the sample surface are facilitated, resulting in an accelerated corrosion process. The behaviors of the as-cast and heat-treated samples were quite similar except for the potential values in the heat-treated samples being slightly higher than those in the as-cast samples, providing an indication of the improved corrosion resistance of the alloy after annealing, as described earlier. Consistent with the observations in Section 3.1, reduction in the fraction of the β phase after annealing reduced the fraction of less noble anodic regions in the microstructure of C95800 and improved the overall corrosion performance of the samples.

For both as-cast and samples heat-treated at 25 and 40 °C solution temperatures, the maximum change in the OCP was observed when the rotation speed was 1000 rpm. Once

the sample was exposed to a mildly flowing corrosive solution (in this case, induced by rotation at 1000 rpm), the rate of electrochemical interactions as well as the properties of the diffuse layer changed dramatically. The properties of the diffuse layer are important at the electrode/solution interface and affect the cathodic or anodic reaction rates. Both the cationic corrosion products and anionic corrosive stimulators should pass the diffuse layer; thus, the diffusivity of the cations and anions in the diffuse layer controls the overall corrosion reactions [27].

Although the trends in the change in the OCP showed the overall effects of heat treatment, temperature, and rotation speed on the corrosion resistance of the C95800 alloy, the quantification of these parameters using OCP results is not feasible. For this reason, further PDP and EIS measurements were performed on as-cast and heat-treated samples at different rotation speeds and different solution temperatures, which is discussed in the following sections.

### 3.2.2. Potentiodynamic Polarization Measurements

Figure 4 provides the PDP results for the as-cast and heat-treated C95800 alloy in 3.5 wt% NaCl at two different solution temperatures (i.e., 25 and 40 °C) under the stagnant condition and under RDE rotation speeds of 1000–9000 rpm. The potential range in the PDP measurements was −200 to +500 mV with respect to the OCP values and the scan rate was set to 1 mV s$^{-1}$. The corrosion current density ($i_{corr}$), corrosion potential ($E_{corr}$), polarization resistance ($R_p$), and corrosion rate ($C_{rate}$) were calculated using the Tafel extrapolation method and model analysis using Ivium Software, and the results are reported in Table 2. As shown in Figure 4a–d and Table 2, for both as-cast and heat-treated samples at 25 and 40 °C solution temperatures, the minimum and maximum corrosion current densities are related to the stagnant and maximum rotation speed (i.e., 9000 rpm) conditions, respectively. In all cases, $E_{corr}$ shifted toward the negative potentials as the sample rotation speed increased, which indicated the detrimental effect of the fluid flow (or sample rotation in this case) on the corrosion resistance of C95800 alloy. Furthermore, the positive impact of the heat treatment on the corrosion resistance of the C95800 alloy was evidenced from the PDP results.

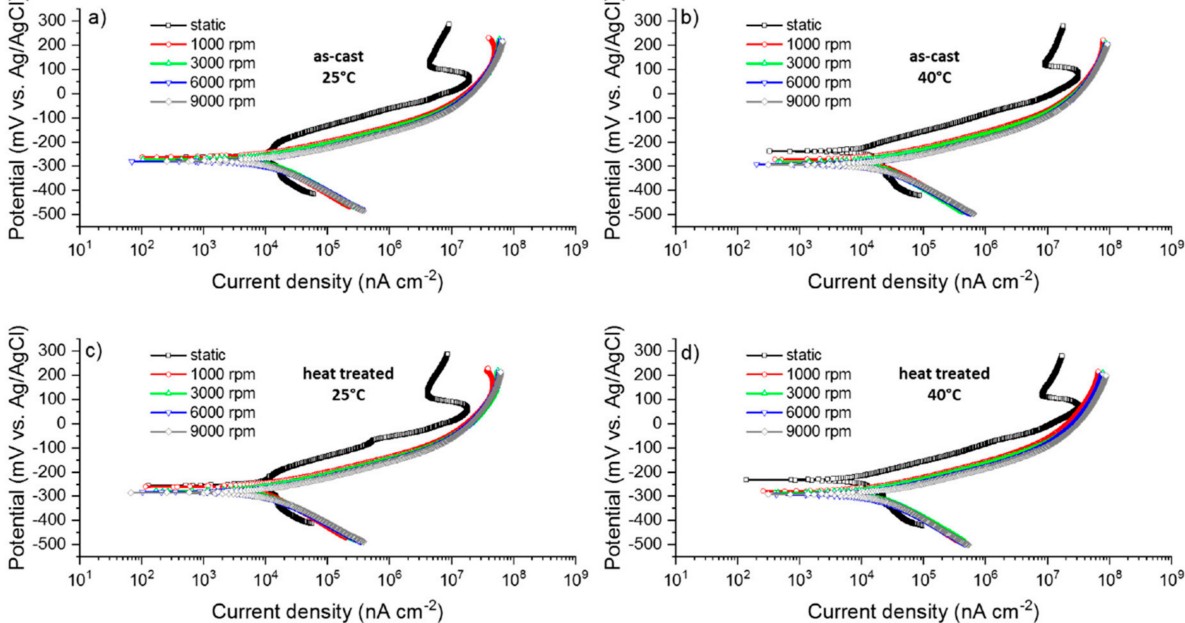

**Figure 4.** Potentiodynamic polarization results of C95800 alloy for (**a**) as-cast sample at 25 °C, (**b**) as-cast sample at 40 °C, (**c**) heat-treated sample at 25 °C, and (**d**) heat-treated sample at 40 °C.

**Table 2.** The obtained electrochemical parameters from the potentiodynamic polarization results in Figure 4.

| Parameter | T (°C) | Static | 1000 rpm | 3000 rpm | 6000 rpm | 9000 rpm | Type |
|---|---|---|---|---|---|---|---|
| E$_{corr}$ (mV) | 25 | −262 | −264 | −270 | −279 | −278 | as-cast |
| | | −255 | −260 | −278 | −282 | −284 | heat-treated |
| | 40 | −237 | −271 | −283 | −291 | −294 | as-cast |
| | | −232 | −278 | −285 | −293 | −292 | heat-treated |
| i$_{corr}$ (A cm$^{-2}$) | 25 | $1.74 \times 10^{-6}$ | $4.95 \times 10^{-6}$ | $5.06 \times 10^{-6}$ | $5.23 \times 10^{-6}$ | $5.57 \times 10^{-6}$ | as-cast |
| | | $9.18 \times 10^{-7}$ | $4.50 \times 10^{-6}$ | $4.86 \times 10^{-6}$ | $5.03 \times 10^{-6}$ | $5.22 \times 10^{-6}$ | heat-treated |
| | 40 | $3.37 \times 10^{-6}$ | $6.61 \times 10^{-6}$ | $7.45 \times 10^{-6}$ | $8.44 \times 10^{-6}$ | $8.97 \times 10^{-6}$ | as-cast |
| | | $1.07 \times 10^{-6}$ | $5.97 \times 10^{-6}$ | $6.7 \times 10^{-6}$ | $7.66 \times 10^{-6}$ | $8.83 \times 10^{-6}$ | heat-treated |
| R$_p$ (Ω cm$^2$) | 25 | - | 1669.70 | 1646.15 | 1593.94 | 1482.08 | as-cast |
| | | - | 1968.39 | 1945.78 | 1879.88 | 1739.95 | heat-treated |
| | 40 | - | 1126.48 | 1056.81 | 936.31 | 887.84 | as-cast |
| | | - | 1373.75 | 1177.89 | 1130.01 | 930.42 | heat-treated |
| C. Rate (mm year$^{-1}$) | 25 | 0.02 | 0.05 | 0.05 | 0.05 | 0.06 | as-cast |
| | | 0.01 | 0.04 | 0.05 | 0.05 | 0.05 | heat-treated |
| | 40 | 0.03 | 0.07 | 0.07 | 0.08 | 0.09 | as-cast |
| | | 0.03 | 0.06 | 0.07 | 0.08 | 0.09 | heat-treated |

As the solution temperature increased from 25 °C to 40 °C, the corrosion rate of as-cast and heat-treated C95800 alloys increased because of the reduction in the activation energy, the reduced diffusion barriers for reactants toward the sample surface, and the diffusion of corrosion products from the sample toward the solution. In other words, the effect of activation and concentration polarizations reduces with increasing solution temperature [27].

A closer look at the PDP results, especially at the lower electrolyte temperature (i.e., 25 °C), revealed that in stagnant conditions, both cathodic and anodic branches show a concentration polarization behavior close to E$_{corr}$, where the main cathodic reaction is oxygen reduction, as [1,28]:

$$O_2 + 2H_2O + 4e^- \rightarrow 4OH^-.$$

The protective layer that is formed on the samples at high overpotential values consists of Cu$_2$O, and Al and Ni oxides, which reduce the uniform corrosion rate. In the low anodic overpotentials, the mass and ion slow transportation through Cu containing corrosion products, resulting in anodic concentration polarization in this region (i.e., slightly above E$_{corr}$). Kear et al. [29] correlated the anodic polarization of copper in saline solutions to the CuCl$_2^-$ mass transportation on the metal surface. With increasing solution temperature, this diffusion limitation is reduced due to the increase in the diffusion rate of ions. The relatively protective corrosion product layer on the surface of C95800 formed at higher overpotentials when the samples were exposed to higher temperatures (i.e., 275 mV overpotential at 25 °C vs. 300 mV overpotential at 40 °C).

In the PDP results of heat-treated samples under the stagnant condition, small but reproducible current fluctuations were observed at 200–250 mV anodic overpotentials, which indicated the formation of corrosion product films, although quite unstable, due to the positive effect of the reduction in the sensitive β phase fraction in the microstructure of heat-treated alloys. At higher anodic overpotentials, under stagnant conditions, a reduction in current density was observed, which was correlated with the formation of a protective layer, presumably in the form of mixed porous and metastable corrosion products, on the

alloy's surface. Further increase in the overpotential, however, resulted in the dissolution of this metastable layer, which manifested in the increase in the current density.

The PDP behavior of the as-cast and heat-treated samples under hydrodynamic flow (i.e., rotating disc) was different from that of the samples in the stagnant condition. The hydrodynamic flow of the solution close to the metal surface facilitated the diffusion of electronegative and small $Cl^-$ ions through the pores of the corrosion product layer and further destabilized it. As observed in the PDP results, the flowing condition mainly affected the anodic branch of the polarization curve, indicating that solution flow facilitates the diffusion and transportation of metal ions.

It has been well-demonstrated that anodic dissolution of Cu is the main reaction in the corrosion of NABs, which results in the formation of $CuCl_2^-$ in the following reaction [1,29,30]:

$$Cu + 2Cl^- \rightarrow CuCl_2^- + e^-.$$

In the presence of $Cl^-$ ions, copper oxides are proposed to form [29,31]:

$$2CuCl_2^- + 2OH^- \leftrightarrow Cu_2O + H_2O + 4Cl^-.$$

Alternatively, the dissolution of CuCl in the form of Cu(I) in a $Cl^-$-containing solution may result in the formation of $Cu_2O$, as [30]:

$$2CuCl + H_2O \leftrightarrow Cu_2O + 2H^+ + 2Cl^-.$$

The stability of $Cu_2O$ is highly dependent on the concentration of $Cl^-$ ions. As described earlier, as the samples were exposed to the saline solution, an oxide layer formed on their surface, in which oxide and chloride complexes precipitated and offered a corrosion barrier layer to the surface [1,32]. This relatively protective layer reduces the rate of anodic dissolution in the growing $Cu_2O$ layer and simultaneously blocks the ions' transportation across the surface layer, thus affecting the cathodic reaction rates [12].

Consistent with the results in Figure 3, as shown in the PDP results, sample rotation up to 1000 rpm led to the most significant changes in the corrosion behavior of the samples compared to the stagnant exposure condition, and further increases in the rotation speed (up to 9000 rpm) only produced minor changes in the PDP results. Furthermore, the behavior of the anodic and cathodic branches of the rotating samples were quite similar, irrespective of their rotation speed. The kinetics of electrochemical reactions is determined by the slowest step (i.e., the rate-determining step). The increase in the accessibility of the surface to oxygen and ions under hydrodynamic flow (i.e., rotation of the sample) as well as the facilitated transportation of corrosion products toward electrolyte promote the rate of anodic reactions [33]. Specifically, as the electrodes start to rotate, induced convection flow increases the dissolution and diffusion of metal ions from the corroding surface toward the solution.

The increase in the metal ion transportation rate coincides with the reduction in the thickness of the diffused layer ($\delta$), which further facilitates ion diffusion toward the metal surface and thus increases the corrosion rate. The thickness of the diffused layer can be determined based on the angular rotation speed ($\omega$), kinematic viscosity ($\nu$), and diffusion coefficient (D) as [27]:

$$\delta = \frac{D^{\frac{1}{3}}}{0.62\omega^{\frac{1}{2}}\nu^{\frac{-1}{6}}}. \tag{1}$$

The changes in $\delta$ as a function of rotation speed at 25 and 40 °C electrolyte temperature are plotted in Figure 5. In this figure, the main changes in $\delta$ appear as the sample exposure condition changes from stagnant to rotating (fluid flow); further increases in the electrode rotation speed change $\delta$. These observations are consistent with the OCP, PDP, and EIS results, where the main changes were observed as the sample started rotating.

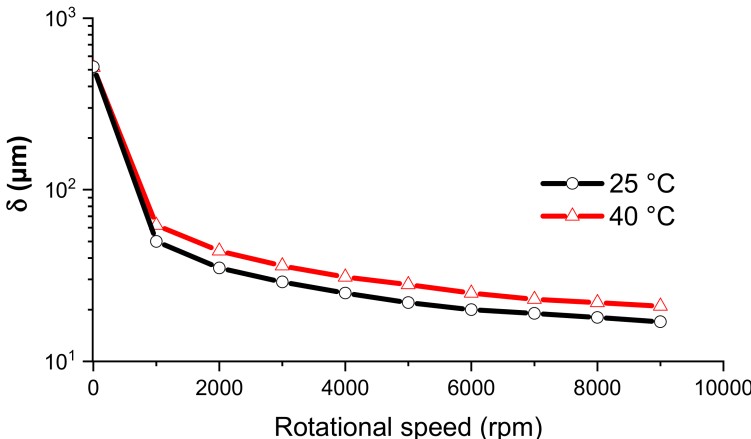

**Figure 5.** Changes in the thickness of the diffused layer as a function of electrode rotation speed at 25 and 40 °C electrolyte temperature.

Besides the abovementioned effects, sample rotation induced a mechano-rheological impact on the surface through the induced shear stress. At fast enough rotation speeds, the cavitation effect can also be observed, which is caused by the burst of the air bubbles at the surface of the solid. Cavitation is especially an issue in rotating parts in seawater, such as propellers [12]. The flow pattern in the RDE cell was proposed by Gonzalez et al. [34], who showed that close to the electrode surface, the fluid flow pattern is asymmetrical, resulting in substantial shear stress on the surface. The magnitude of this shear stress depends on the rheological properties of the fluid as well as the rotation speed. Once this induced shear stress surpasses the bonding strength between the corrosion products and the metal surface, in a rotating sample, this protective layer is partially or completely damaged. The damaged corrosion product film is less protective than the undamaged one; thus, the corrosion rate in the rotation samples increases further due to this mechanical damage. Figure 6 depicts the calculated induced shear stress as a function of electrode rotation speeds under different electrolyte temperatures (25 and 40 °C). The difference between the shear stress at 25 and 40 °C electrolyte temperatures occurred due to the impact of temperature on the viscoelastic properties of water as the main constituent of the electrolyte [35]. Based on the maximum stress level observed in this figure, the electrochemical reactions are dominant in the degradation of C95800 up to 9000 rpm.

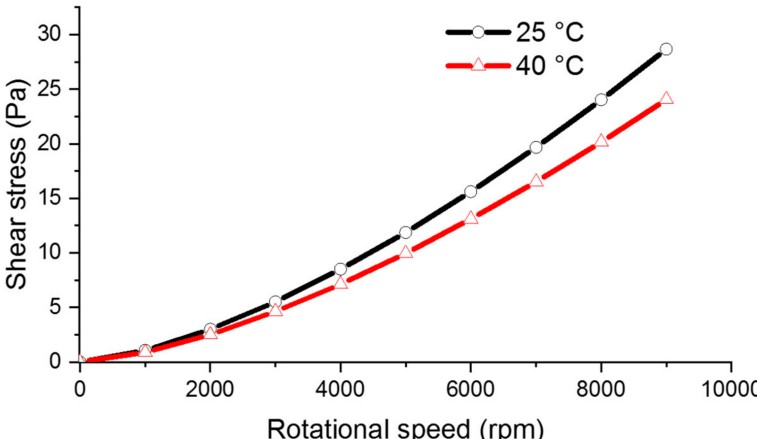

**Figure 6.** The magnitude of the shear stress as a function of electrode rotation speed at 25 and 40 °C electrolyte temperatures.

### 3.2.3. Electrochemical Impedance (EIS) Measurements

To further assess the mechanism of corrosion as well as the properties of the protective layer on as-cast and heat-treated C95800 alloys under stagnant and rotating conditions, complementary EIS measurements were performed. The EIS measurements were performed directly after the OCP measurement, in which the steady-state condition was achieved after 1000 s of exposure to the solution. The EIS measurements were performed in the frequency range of $10^{-2}$ to $10^5$ Hz with the amplitude of perturbation potential of $\pm 10$ mV. The Nyquist representation of the EIS results for as-cast and heat-treated samples is provided in Figure 7. Bode phase diagrams showing the effect of rotation speed on the corrosion behavior of the heat-treated samples at the 25 °C electrolyte temperature are provided in Figure 8a. Representative Bode phase diagrams corresponding to as-cast and heat-treated samples under 1000 rpm rotation condition at 25 and 40 °C electrolyte temperatures are presented in Figure 8b.

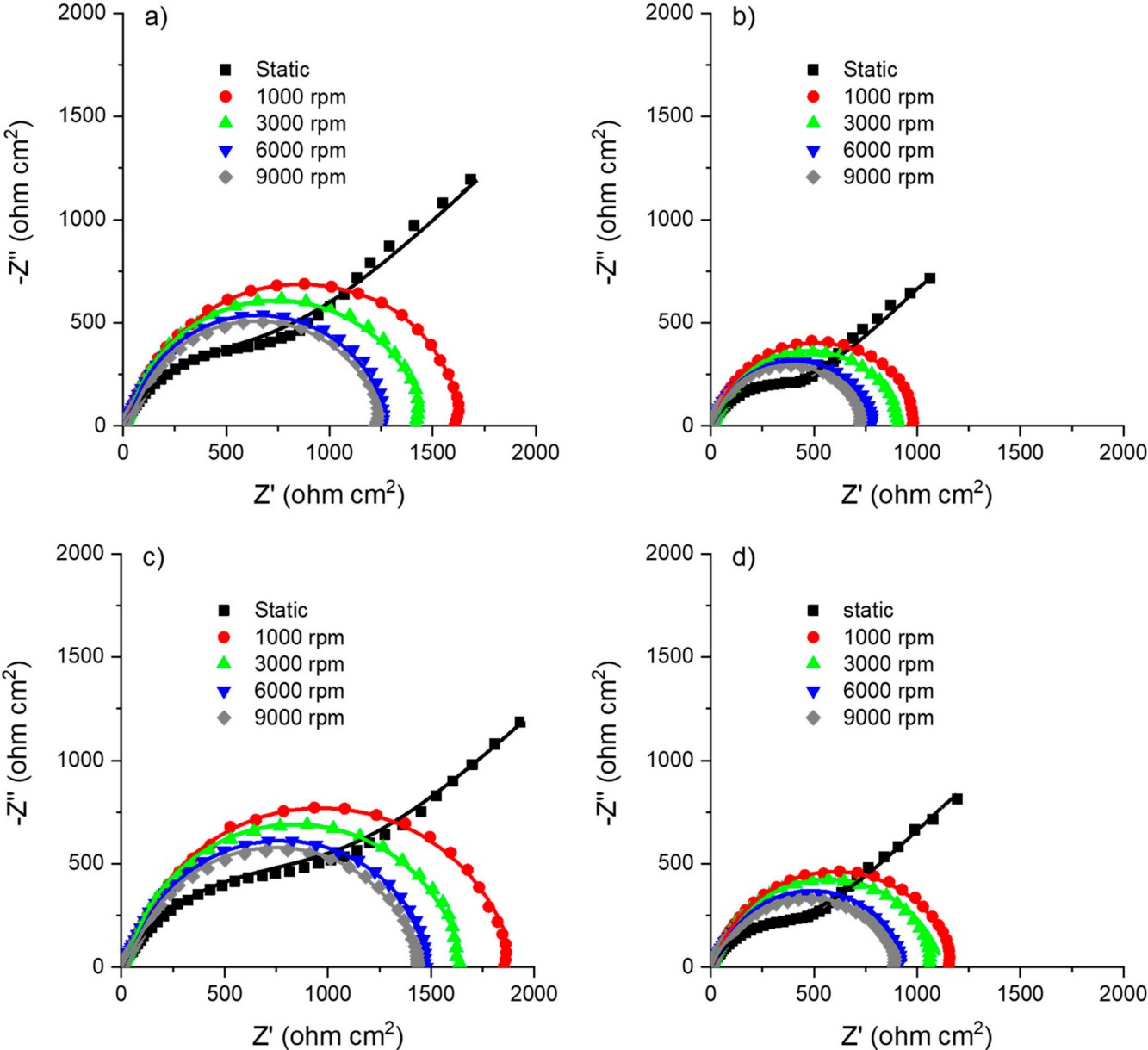

**Figure 7.** Nyquist representation of the electrochemical impedance spectroscopy results of C95800 alloy for (**a**) as-cast sample at 25 °C, (**b**) as-cast sample at 40 °C, (**c**) heat-treated sample at 25 °C, and (**d**) heat-treated sample at 40 °C.

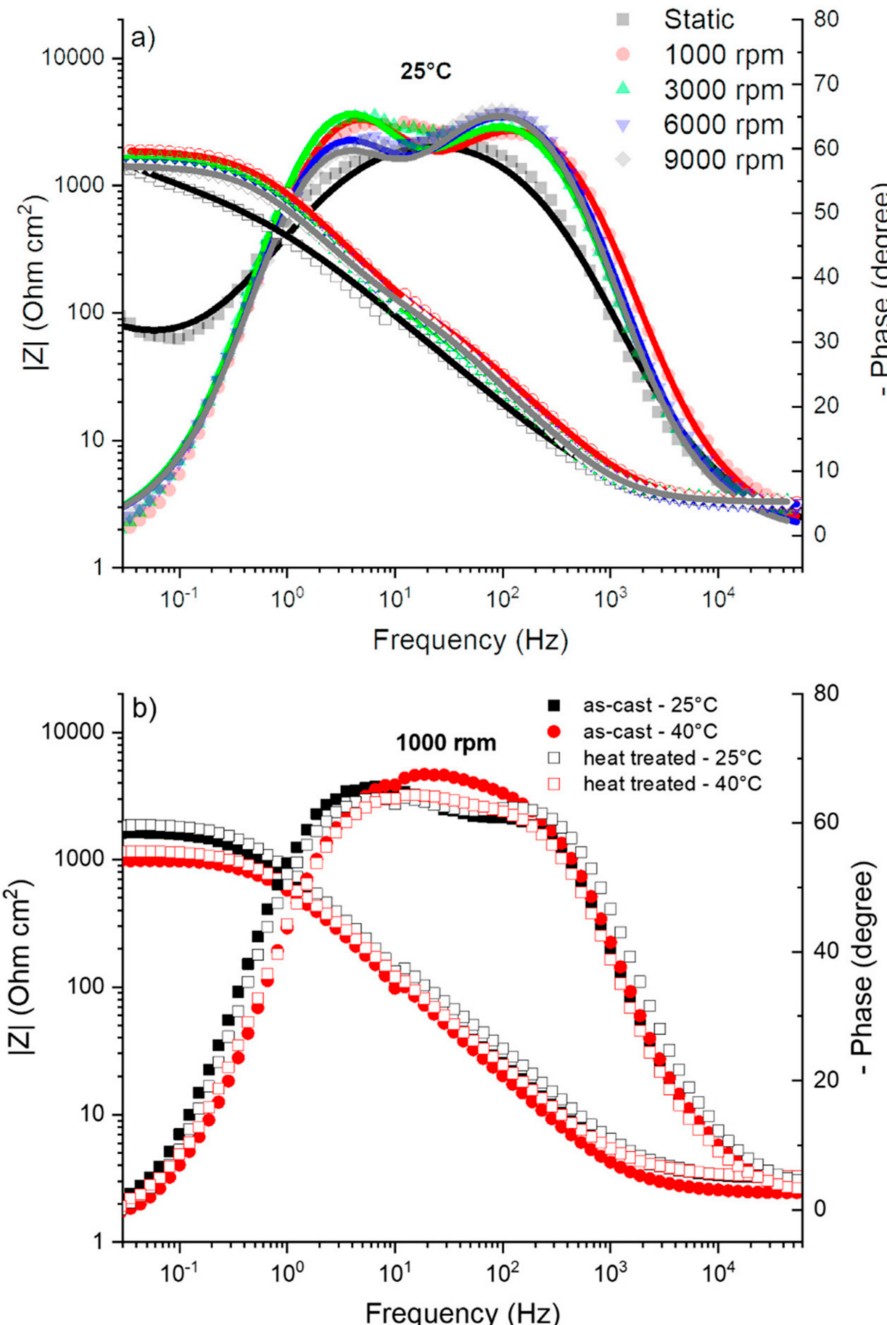

**Figure 8.** (**a**) Bode phase diagram showing the effect of rotation speed on the corrosion behavior of samples heat-treated with a 25 °C electrolyte temperature. Continuous lines are fit to the experimental data points. (**b**) Representative Bode phase diagram showing the results for as-cast and heat-treated samples under a 1000 rpm rotation speed at 25 and 40 °C electrolyte temperature.

As shown in Figure 7 and Table 3, the total impedance values of the heat-treated samples were always higher than those of the as-cast samples in identical conditions, which confirmed the positive impact of heat treatment on the corrosion resistance of the C95800 alloy in both stagnant and rotating conditions, consistent with our previous results (Vide Supra). The detrimental effect of the electrode rotation (fluid flow) on the corrosion resistance of alloys is clear based on these results. In the EIS results, the change in the corrosion behavior of the C95800 alloy in the stagnant exposure condition and under rotation was very clear (Figures 7 and 8). The EIS response of the as-cast and heat-treated samples under stagnant exposure conditions at both temperatures exhibited the formation of a porous

surface layer (represented by the Warburg element in the equivalent circuit), which disappeared as the sample rotated at 1000 rpm. The presence of Warburg behavior in the EIS response of the samples under the stagnant condition confirmed that the electrochemical interactions at the metal/electrolyte interface are controlled by the transportation of metal ions and oxygen. In this case, the corrosion resistance of the sample would be sensitive to electrolyte flow, which changes the corrosion-controlling mechanism to mixed polarization. Rotation of the sample (or fluid flow) would facilitate mass transportation and thus reduce the concentration polarization and corrosion resistance of the samples. Further increases in the sample rotation speed only affect the severity of electrochemical interactions, without a clear change in the corrosion mechanism. Accordingly, different equivalent circuit models were used to fit the EIS experimental data, as presented in Figure 9, and the corresponding fitting parameters are provided in Table 3.

**Table 3.** Parameters obtained by fitting the EIS results using the equivalent circuit models presented in Figure 9.

| Parameter | T (°C) | Static | 1000 rpm | 3000 rpm | 6000 rpm | 9000 rpm | Type |
|---|---|---|---|---|---|---|---|
| $R_s$ ($\Omega$ cm$^2$) | 25 | 3.25 | 3.17 | 3.19 | 3.18 | 3.26 | as-cast |
| | | 3.01 | 3.15 | 3.25 | 3.12 | 3.01 | heat-treated |
| | 40 | 2.35 | 2.42 | 2.41 | 2.35 | 2.35 | as-cast |
| | | 2.33 | 3.25 | 2.69 | 2.76 | 2.32 | heat-treated |
| $R_{oxide}$ ($\Omega$ cm$^2$) | 25 | - | 159.81 | 181.11 | 207.20 | 205.06 | as-cast |
| | | - | 220.25 | 195.76 | 212.54 | 217.33 | heat-treated |
| | 40 | - | 244.74 | 219.68 | 315.39 | 329.39 | as-cast |
| | | - | 203.21 | 302.00 | 334.07 | 363.18 | heat-treated |
| $R_{ct}$ ($\Omega$) | 25 | 838.65 | 1507.87 | 1290.50 | 1079.27 | 1053.08 | as-cast |
| | | 902.16 | 1686.78 | 1460.14 | 1288.93 | 1224.40 | heat-treated |
| | 40 | 458.85 | 768.47 | 705.95 | 482.22 | 411.30 | as-cast |
| | | 494.63 | 984.84 | 743.68 | 614.94 | 533.73 | heat-treated |
| $C_{oxide}$ (F) | 25 | - | $6.74 \times 10^{-6}$ | $6.55 \times 10^{-6}$ | $5.89 \times 10^{-6}$ | $6.5 \times 10^{-6}$ | as-cast |
| | | - | $5.26 \times 10^{-6}$ | $6.08 \times 10^{-6}$ | $5.5 \times 10^{-6}$ | $5.46 \times 10^{-6}$ | heat-treated |
| | 40 | - | $8.64 \times 10^{-6}$ | $6.4 \times 10^{-6}$ | $6.42 \times 10^{-6}$ | $6.5 \times 10^{-6}$ | as-cast |
| | | - | $7.48 \times 10^{-6}$ | $5.46 \times 10^{-6}$ | $5.53 \times 10^{-6}$ | $5.14 \times 10^{-6}$ | heat-treated |
| $n_1$ | 25 | 0.74 | 0.84 | 0.85 | 0.86 | 0.84 | as-cast |
| | | 0.73 | 0.84 | 0.85 | 0.86 | 0.87 | heat-treated |
| | 40 | 0.74 | 0.84 | 0.83 | 0.84 | 0.84 | as-cast |
| | | 0.73 | 0.83 | 0.84 | 0.83 | 0.84 | heat-treated |
| $C_{dl}$ (F) | 25 | $2.01 \times 10^{-5}$ | $2.94 \times 10^{-6}$ | $2.61 \times 10^{-6}$ | $2.81 \times 10^{-6}$ | $2.01 \times 10^{-5}$ | as-cast |
| | | $2.01 \times 10^{-5}$ | $2.57 \times 10^{-6}$ | $3.22 \times 10^{-6}$ | $4.75 \times 10^{-6}$ | $2.01 \times 10^{-5}$ | heat-treated |
| | 40 | $2.25 \times 10^{-5}$ | $2.03 \times 10^{-6}$ | $2.67 \times 10^{-6}$ | $1.87 \times 10^{-6}$ | $2.25 \times 10^{-5}$ | as-cast |
| | | $2.22 \times 10^{-5}$ | $2.47 \times 10^{-6}$ | $3.61 \times 10^{-6}$ | $2.47 \times 10^{-6}$ | $2.22 \times 10^{-5}$ | heat-treated |
| $n_2$ | 25 | - | 0.97 | 0.99 | 0.99 | 0.96 | as-cast |
| | | - | 0.96 | 0.98 | 0.93 | 0.90 | heat-treated |
| | 40 | - | 0.99 | 0.97 | 0.99 | 0.99 | as-cast |
| | | - | 0.92 | 0.85 | 0.97 | 0.87 | heat-treated |

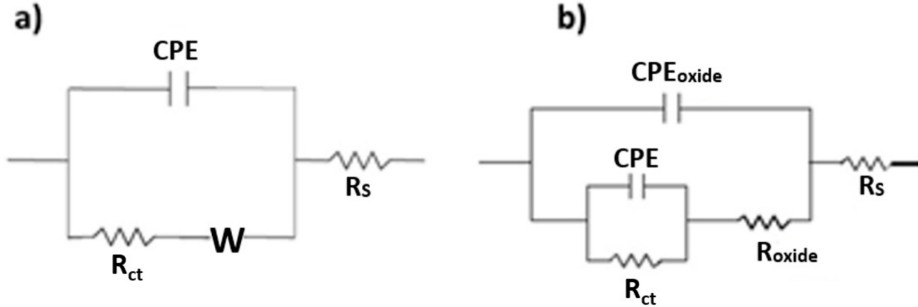

**Figure 9.** The equivalent circuit models used for fitting the electrochemical impedance spectroscopy results for the (**a**) stagnant condition and (**b**) rotating condition.

In the equivalent circuit models, $R_s$ represents the solution resistance, $R_{ct}$ is the charge transfer resistance at the interface between the metal and electrolyte, $R_{oxide}$ is the resistance of the surface layer (oxide), W is the Warburg element, and $C_{dl}$ and $C_{oxide}$ correspond to the capacitance of the EDL (electric double layer) and surface film, respectively. In the fitting of the EIS results, due to the heterogeneity of the rough surface layer, constant phase elements (CPE and CPE$_{oxide}$) were used instead of an ideal capacitor [36], in which the $C_{dl}$ and $C_{oxide}$ values were calculated based on the following equations:

$$Z_{CPE} = \frac{1}{Q(j\omega)^n},$$ (2)

$$C(\text{oxide or } C_{dl}) = Q^{\frac{1}{n}} R^{\frac{1-n}{n}},$$ (3)

where Q and n are the characteristic parameters of the CPE: Q as a numerical value of the admittance $(1/|Z|)$ at $\omega = 1$ rad s$^{-1}$ with unit of S·s$^n$, and n as an indication of surface heterogeneity factor (n = 1 for an ideal flat capacitance), which is dimensionless. The corresponding values are listed in Table 3.

The observed decrease in $R_{ct}$ with increasing the rotation speed from 1000 to 9000 rpm is consistent with the PDP results, indicating the reduction in the corrosion resistance of the samples with increasing rotation speed. Slight variations in $R_{oxide}$ seem to be related to the changes in surface oxide layer properties by increasing the rotation speed from 1000 to 9000 rpm. Nevertheless, the simultaneous electrochemical and hydrodynamic effects that contribute to the changes of corrosion of samples under flowing conditions do not allow for a straightforward description of the corroding systems based on a single parameter, and the total polarization resistance ($R_P$) should be considered as $R_P = R_{ct} + R_{oxide}$ to include the effect of sample rotation on the overall corrosion resistance of samples.

In the Bode phase results in Figure 8, two time constants can be recognized for rotating samples: one corresponding to the activation polarization and capacitance of the charge transfer layer (at low frequencies) and the other corresponding to the resistance of the surface layer (oxide) and its capacitance (at higher frequencies). The positive impact of heat treatment and the effect of solution temperature are also shown in both Nyquist and Bode phase plots; with increasing the electrolyte temperature from 25 to 40 °C, $R_s$ reduced in all cases, which confirms the increase in the ionic conductivity of the solution and increased diffusion rate with increasing temperature. Consequently, corrosion resistance decreases as the electrolyte temperature increases.

In RDE measurements, the correlation between corrosion current density and the rotation velocity indicates the types of corrosion reaction mechanisms [27]. The linear increase in corrosion current density versus disc rotation rate represents a concentration polarization or diffusion-controlled reaction, whereas under the activation-controlled corrosion mechanism, the corrosion current density is independent of the disc rotation rate. Figure 10 depicts the correlation between inverse polarization resistance ($R_P^{-1}$) and the square root of the angular frequency of rotation of the disc ($\sqrt{\omega}$) for as-cast and heat-treated

samples at 25 and 40 °C electrolyte temperatures. As shown in this figure and consistent with the previous results, the corrosion resistance for both as-cast and heat-treated samples decreased as the electrolyte temperature increased from 25 to 40 °C. Increasing the disc rotation speed increased the corrosion rate. In all cases, the corrosion followed the Levich model [37]; thus, the corrosion reaction was under control of the diffusion of metal ions from the sample surface to the solution. Furthermore, the positive effect of heat treatment on the corrosion resistance of C95800 alloy was more significant at higher electrolyte temperatures and higher disc rotation rates.

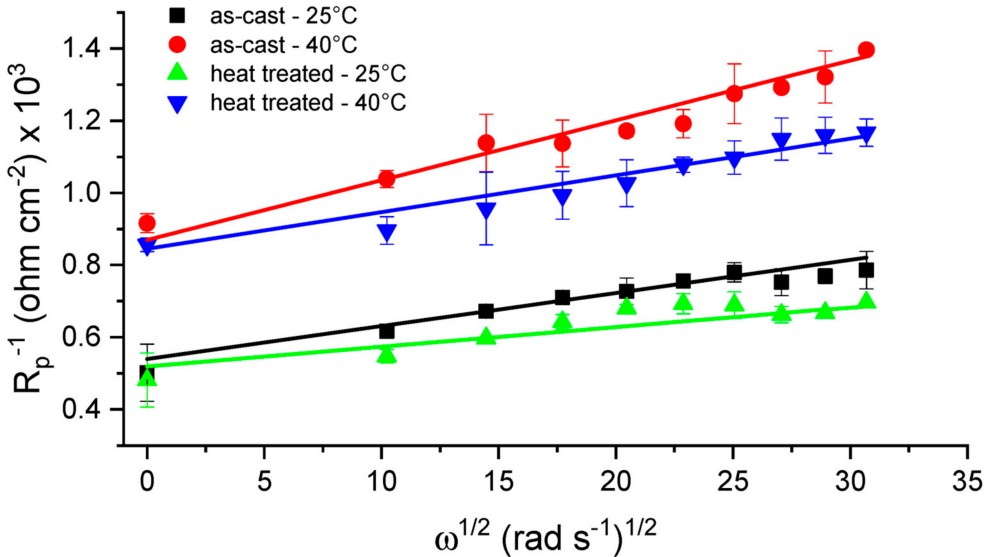

**Figure 10.** The correlation between inverse polarization resistance and the square root of sample rotation speed for as-cast and heat-treated samples in 25 and 40 °C electrolyte temperatures. The continuous lines are linear fits to the experimental data.

### 3.3. Macroscopic Optical Observations

The macroscopic characteristics of RDE samples were evaluated after PDP measurements and the results are presented in Figure 11. In this figure, the effect are clear of disc rotation rate and its impacts on the severity of the electrochemical and corrosion reactions on the surface of electrodes. In particular, at electrode rotation rates above 2000 rpm, two separate regions can be observed on the sample surface, which are separated by a narrow boundary. The central region increased in size as the rotation rate increased and became darker and rougher. Therefore, the anodic reactions appeared more pronounced in this central area. This behavior can be explained by the difference in the linear speed between central regions of the disc and those closer to its circumference. In a planar rotating disc, the centrifugal force generates a flow pattern of the liquid layer close to the surface toward the perimeter of the disc, whereas the bulk electrolyte is forced toward the center of the disc. Meanwhile, the perimeter of the disc experiences the most rapid linear flow and shear stress, which increases with increasing rotation speed.

Therefore, the patterns of attack must be related to shear stresses acting on the protective layers and the enhanced transportation of metal ions, both of which are controlled by the flow rate and are accelerated in an uncontrolled fashion by the roughening of the surface during the experiment. Consequently, the surface area of the central damaged region extended as the rotation speed increased.

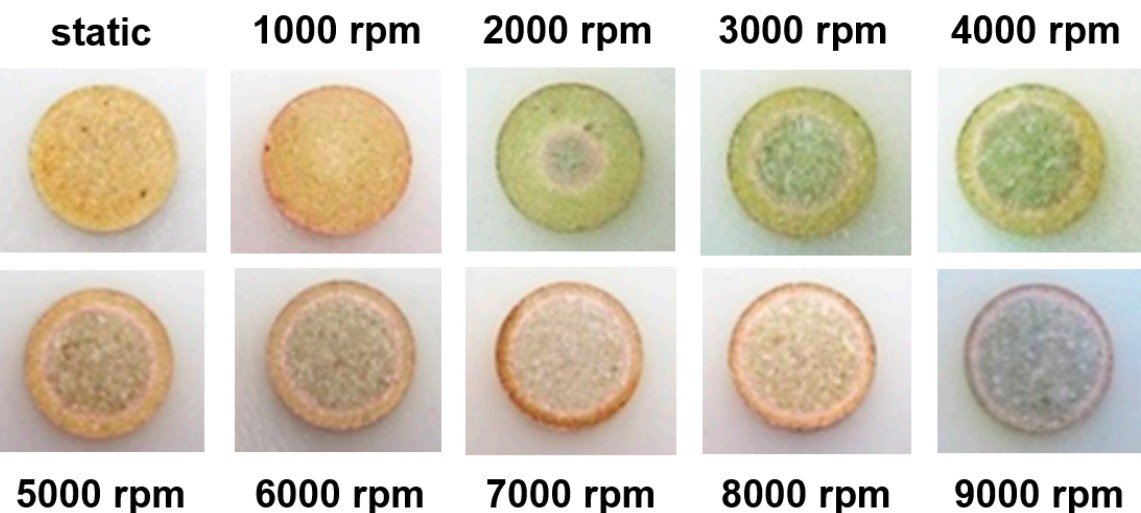

**Figure 11.** Surface of rotating disc electrodes after potentiodynamic polarization measurement at different electrode rotation speeds (the diameter of samples is 5 mm).

## 4. Conclusions

The corrosion resistance of C95800 NAB alloy was investigated under stagnant and hydrodynamic flow conditions using an electrolyte simulated by RDE in a saline solution. We observed that changing the exposure condition from stagnant electrolyte to flow significantly impacted the corrosion behavior of the studied samples. With increasing RDE rotation rate, the anodic reactions and metal dissolution rate close to the sample surface increased whereas the thickness of the diffused layer slightly decreased. Meanwhile, the magnitude of shear stress close to the sample surface increased with sample rotation rate. Increasing the flow rate (i.e., disc rotation rate) and increasing the electrolyte temperature both reduced the corrosion resistance of the as-cast C95800 samples. After annealing heat treatment of the alloy at 675 °C for 6 h, the corrosion resistance of C95800 was improved in stagnant as well as hydrodynamic flow conditions, a result that was correlated with the decrease in the phase fraction of corrosion-sensitive phase β in the microstructure of C95800 after heat treatment. The positive effect of annealing heat treatment on the corrosion resistance of alloy was more pronounced at higher electrolyte temperature and faster RDE rotation rates.

**Author Contributions:** Conceptualization, H.R.J., A.D. and S.H.; methodology, H.R.J., A.D. and S.H.; formal analysis, H.R.J., A.D. and S.H.; investigation, H.R.J.; resources, A.D. and S.H.; writing—original draft preparation, H.R.J., A.D. and S.H.; writing—review and editing, A.D. and S.H.; visualization, S.H.; supervision, A.D. and S.H.; project administration, A.D.; funding acquisition, A.D. All authors have read and agreed to the published version of the manuscript.

**Funding:** Authors declare no funding for this research.

**Data Availability Statement:** The data presented in this study are available on request from the corresponding author. The data are not publicly available until the publication of the follow up papers.

**Acknowledgments:** H.R.J. acknowledges New Materials Technology and Processing Research Center at Neyshabur Branch, Islamic Azad University for financial supporting. S.H. thanks W. Peukert and Emerging Talents Initiative (ETI) 2018/2_Tech_06, FAU, Germany, and BMBF-MSRT (grant ID: CAlSAB) for supporting his research.

**Conflicts of Interest:** The authors declare no conflict of interest.

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
