# Peer review of "Effect of Fluid Flow on the Corrosion Performance of as-Cast and Heat-Treated Nickel Aluminum Bronze Alloy (UNS C95800) in Saline Solution"

_cmd, doi:10.3390/cmd2010004_

Round 1

Reviewer 1 Report

The paper addresses the effect of a heat treatment on the corrosion behaviour of NAB under flow conditions in saline environment. This topic is not new but of continuing interest as this material is frequently used under high flow conditions, particularly in seawater. Electrochemical methods are used to asses the corrosion behaviour.

The paper is well written, clear, and the over all conclusion is supported by the data: The heat treated material appears better corrosion resistant compared to the as cast material used in this study. Just a few typing mistakes were encontered, see below.

However, the interpretation of the data and important explanations are dubious and sometimes fundamentally(!) wrong. In particular the role of oxygen diffusion is heavily misinterpreted and overestimated. See below.

Consequently,rejecting the manuscript is proposed.

The data and final conclusion related to material behaviour deserve publication, but the interpretation must be adapted to fit the data without overinterpretation. Please consult a corrosion-experienced electrochemist for proper interpretation. I am looking forward seeing this research being published properly!

--------------------------------------------------------
typing mistakes (occasionally identified, list not exhaustive):
line 109: "reach" should be rich
line 118: "showed" should be shown
line 188, 226: "mv" should be mV
--------------------------------------------------------

Abstract
line 25: ...the corrosion mechanism is under oxygen diffusion control....
This is an irritating statement!
The authors want investigate the corrosion behaviour of 2 different materials (as cast; heat treated).
If the corrosion test was controlled by oxygen diffusion in both cases, this implies that the experimental conditions were unsuitable to discriminate any difference in reactivity of the materials.
I.e. the abstract is contradicting itself.

2.3 and/or 3.2.2
the potential scan rate is missing.

3.1
how were the phases quantified?

line 139-141
This sentence is highly dubious, here appears a/the fundamental error in interpretation!
- at OCP, nothing is polarized
- the potentials are shifted to more negative values by flow. This indicates the major impact of flow being the acceleration of the anodic reaction !!!!!
Explanation: If you consider OCP being the potential where anodic and cathodic partial currents have the same intensity, and if you consider that flow will increase the slope of both in an I/E diagram, you will see that a shift of OCP to negative can only be explained by the effect of flow being much stronger on the anodic reaction than on the cathodic reaction.
This is applied textbook knowledge about I/E diagrams, you may also find it in the standard ISO 8044, Figure 1.
CONCLUSION: The effect of flow is enhancing the diffusion of the metal ions from the surface. The effect on the diffusion of oxygen is much less, i.e. oxygen diffusion is NOT the limiting process.

Figure 4
This figure clearly shows that flow has a strong impact on the anodic reaction, the anodic branches under flow are substantially offset from the branch under stagnation, to much higher values.
In contrast, the short cathodic branch is not very much affected by flow, because there is no diffusion limitation for oxygen close to OCP.
CONCLUSION: Flow affects the anodic reaction, i.e. metal dissolution, by enhancing the diffusion/removal of metal ions.

Table 2
How was the Tafel-method done exactly? (software? or? by hand?) - the results are doubtful
icorr is lower under static conditions compared to flow (this is plausible)
Rp is much lower under static conditions than under flow --> this contradicts the data of icorr and is not plausible here.
This could be an artefact from software or its use.
moreover:
- there are mistakes in the Ecorr values (static/25C/heat-treated) and (static/40C/heat-treated)

lines 259-281, Table 3
This section is obsolete, since the diffusion of oxygen to the rotating disc electrode is not related to the material and does not tell anything about the corrosion behaviour of the metal.

Figure 8 / table 4
The fitted data deviate substantially from the measured values at the low frequency end, which is the important end for corrosion rates. In particular figure 8c creates severe doubts on the fitting quality.
Consequently, the equivalent circuit Figure 10b is wrong and/or the fitting routine was not working/ was not used properly. Which software was used? where do the equivalent circuits come from?
All this complicated interpreation is not really necessary: Figure 8 clearly shows that the heat treated material is somewhat better compared to the as cast: The diameters of the semicircles are wider for the heat treated material.
All interpretation beyond this simple approach requires skilled analysis of the impedance data with accurate fitting.

line 356
the term "EDL" is undefined

line 367-368
it says "sample rotation......increases Cdl" (d in the formula is undefined!)
This sounds plausible since the thickness should decrease, but definitely contradicts the data in table 4!

lines 373-376
NO. In Figure 8 we do not see an additional time constant in the high frequency range, but we see a time constant in the low frequency range, which is obviously missing in the model, since not fitted correctly (figure 8c).
Yes, in Figure 9 we see two time constants. Unfortunately, the quality of fitting is not clearly visible in Figure 9.

3.3.
The explanation is highly dubious, because oxygen did not play any role in creating the pattern of corrosion attack! These specimens were subjected to the PDP measurement, i.e. the metal dissolution was driven by the electric current from the potentiostat, and NOT by oxygen.
All the patterns of attack must be related to shear stresses acting on protective layers and enhanced transport of metal ions, all controlled by the flow and additionally accelerated in an uncontrolled way by the roughening of the surface during the experiment.

Figure 12
Why do we see 90% irrelevant background and the really interesting electrode surfaces are so small?

Reviewer 2 Report

The corrosion behaviour of an as-cast and heat treated nickel aluminium bronze casting alloy was studied in contact with 3.5 wt% NaCl solution is investigated under static and dynamic conditions.

The application and the results obtained in NAB alloy have been further explained and the manuscript presents interesting results for the science community working in this topic. I recommend the publication of the manuscript after minor revision.

Minor comments:

  • A picture or scheme of the experimental set-up would be helpful
  • Differences between PDP curves in Figure 4 are not clear. A magnification of the curves at higher frequencies is needed
  • It is not clear the fitting results of the equivalent circuits in Nyquist plots. An error estimation of these values would be interesting

Round 2

Reviewer 1 Report

cmd-1050858-peer-review-v2

The manuscript has improved substantially.

Nevertheless, there are some issues still to be solved before publication (Lxxx = line xxx).

L58: not "Schlüsser" but "Schüssler", see your references

Figure 2 and related text:
Software was used for image analysis. This has caused a highly unrealistic overestimation in quantification of the Kappa-phases.
As we see from Figure 1b,d, kappa are just small dots and fine lines.
The program has substantially overestimated the kappa in relation to alpha/beta by thickening the lines of these fine structures. (see your color coded figures in the author´s response). So, it is an artefact.
This phenomenon could be pointed out in the text and the Figure 2 caption could state eg. "...as estimated by software".
Then it is clear, where these crazy values come from. 
Nevertheless, the relative increase in kappa versus alpha/beta seems correct, it is visible in Figure 1b/d anyway. Therefore the alternative: Omit Figure 2 and the related %values in the text. Instead, refer to Figure 1 in more detail.

L160: ANODIC, not "cathodic" reactions !!!!

L281: Some problem with the Word program (Error! reference not found)

Table 3: 
Cdl - I suppose the unit should be "F", not µF. 
Check also correctness of the unit for Coxide!

L355: Rp = polarization resistance
ok, but it would make sense to deonte it "Rct" (charge transfer resistance) here and in table 3 and figure 9
Because:
L376 says correctly that "Rp+Roxide" should be considered - and this sum is exactly what we usually call the "polarization resistance" (total R at frequency 0) in corrosion science. 
This R(@f=0) is inversely proportional to the corrosion rate. (It should correlate with your Rp from table 2, by the way.)
So, L376 could say "Rp=Rct+Roxide, this is the overall corrosion resistance or polarization resistance, Rp"

L367: Repeat, theoretically Cdl should increase, that´s ok!
But the data in table 3 are not fully consistent with this theoretical expectation. Thus, either omit this section (discussing Cdl is not needed in this work) or rearrange the text so that the relativisation of line 374 applies also for Cdl.

Author Response

Comments and Suggestions for Authors: cmd-1050858-peer-review-v2

The manuscript has improved substantially.

Authors’ response: We thank the reviewer for evaluating our responses to the previous comments positive and thank the reviewer for her/his additional comments on our manuscript, all of which are now implemented in the revision #2. Moreover, some minor spelling and grammatical errors are corrected in the revised manuscript.

Nevertheless, there are some issues still to be solved before publication (Lxxx = line xxx).

L58: not "Schlüsser" but "Schüssler", see your references

Authors’ response: We apologize for the typo and have corrected it in the revised manuscript.

Figure 2 and related text:
Software was used for image analysis. This has caused a highly unrealistic overestimation in quantification of the Kappa-phases.
As we see from Figure 1b,d, kappa are just small dots and fine lines.
The program has substantially overestimated the kappa in relation to alpha/beta by thickening the lines of these fine structures. (see your color coded figures in the author´s response). So, it is an artefact.
This phenomenon could be pointed out in the text and the Figure 2 caption could state eg. "...as estimated by software".
Then it is clear, where these crazy values come from. 
Nevertheless, the relative increase in kappa versus alpha/beta seems correct, it is visible in Figure 1b/d anyway. Therefore the alternative: Omit Figure 2 and the related %values in the text. Instead, refer to Figure 1 in more detail.

Authors’ response: Thanks for pointing this out. As suggested by the reviewer, we have included the following text in the revised manuscript, L 140-143: ”The fraction of Kappa phase in relation with α and β phases may have been overestimated by the software, due to the fine structure of Kappa phase. Nevertheless, the relative increase in the fraction of Kappa phase after heat treatment is clearly visible from Figure 1.”

The following text is also added to the caption of Figure 2: “The phase fractions are estimated using image analysis software.”

L160: ANODIC, not "cathodic" reactions !!!!

Authors’ response: We agree with the reviewer. The text is corrected accordingly.

L281: Some problem with the Word program (Error! reference not found)

Authors’ response: We suppose this error is caused by the mismatch between the word versions, as this error does not appear on the authors version. We have made sure that the pdf version as well as the resubmitted manuscript are free of such errors.

Table 3: 
Cdl - I suppose the unit should be "F", not µF. 
Check also correctness of the unit for Coxide!

Authors’ response: Thanks for pointing these out. Both units are in fact F and are corrected in the revised manuscript.

L355: Rp = polarization resistance
ok, but it would make sense to deonte it "Rct" (charge transfer resistance) here and in table 3 and figure 9
Because:
L376 says correctly that "Rp+Roxide" should be considered - and this sum is exactly what we usually call the "polarization resistance" (total R at frequency 0) in corrosion science. 
This R(@f=0) is inversely proportional to the corrosion rate. (It should correlate with your Rp from table 2, by the way.)
So, L376 could say "Rp=Rct+Roxide, this is the overall corrosion resistance or polarization resistance, Rp"

Authors’ response: We agree with the reviewer and have replaced the Rp with Rct in the text, Table 3, and Figure 9.

L367: Repeat, theoretically Cdl should increase, that´s ok!
But the data in table 3 are not fully consistent with this theoretical expectation. Thus, either omit this section (discussing Cdl is not needed in this work) or rearrange the text so that the relativisation of line 374 applies also for Cdl.

Authors’ response: We agree with the reviewer and have omitted the discussion of the changes of Cdl (i.e. equation 4 is removed in the revised manuscript)

We would like to once more thank the reviewer #1 for careful and critical assessment of our manuscript and for her/his constructive comments. We hope that the revised manuscript is now suitable for publication in CMD.

Reviewer 2 Report

The reviewer comments were addressed and I recommend the paper publication

Author Response

We thank the reviewer for recommending our manuscript for publication in CMD.

Round 3

Reviewer 1 Report

Almost completed!

  • The error in line 284 still persits
  • Line 369: There is still "Rp" but it should be Rct now.
  • Line 376: Rp must be defined in the text as "polarization resistance"; otherwise, the formula is formally out of context, which would be poor practice, in general.

I may recommend careful consideration of such details before submitting a manuscript.

Besides this, I wish seeing the paper published soon in the present form.